# Herpesvirus Replication Compartments: Dynamic Biomolecular Condensates?

**DOI:** 10.3390/v14050960

**Published:** 2022-05-04

**Authors:** Enrico Caragliano, Wolfram Brune, Jens B. Bosse

**Affiliations:** 1Leibniz Institute for Experimental Virology (HPI), 20251 Hamburg, Germany; enrico.caragliano@cssb-hamburg.de; 2Centre for Structural Systems Biology, 22607 Hamburg, Germany; 3Institute of Virology, Hannover Medical School, 30625 Hannover, Germany; 4Cluster of Excellence RESIST (EXC 2155), Hannover Medical School, 30625 Hannover, Germany; 5German Center for Infection Research (DZIF), Partner Site Hamburg-Lübeck-Borstel-Riems, 38124 Braunschweig, Germany

**Keywords:** liquid–liquid phase separation, biomolecular condensate, herpesvirus, CMV, maturation, replication compartment

## Abstract

Recent progress has provided clear evidence that many RNA-viruses form cytoplasmic biomolecular condensates mediated by liquid–liquid phase separation to facilitate their replication. In contrast, seemingly contradictory data exist for herpesviruses, which replicate their DNA genomes in nuclear membrane-less replication compartments (RCs). Here, we review the current literature and comment on nuclear condensate formation by herpesviruses, specifically with regard to RC formation. Based on data obtained with human cytomegalovirus (human herpesvirus 5), we propose that liquid and homogenous early RCs convert into more heterogeneous RCs with complex properties over the course of infection. We highlight how the advent of DNA replication leads to the maturation of these biomolecular condensates, likely by adding an additional DNA scaffold.

## 1. Introduction

Herpesviruses are enveloped, double-stranded DNA viruses with large genomes that replicate in the host cell nucleus. Nine human herpesviruses are currently known, which are highly prevalent in the human population. Herpesviruses persist after primary infection in the host in a state called latency. Reactivation can lead to severe disease in immunocompromised patients, such as transplant recipients, AIDS, or cancer patients [1]. Despite their disease burden, few treatment options exist, with long-term toxicity and the development of viral resistance being a significant concern [2,3]. They are divided into three subfamilies based on biological criteria: *Alphaherpesvirinae* (HSV-1, 2 and VZV), *Betaherpesvirinae* (HCMV, HHV-6A and B, and HHV-7) and *Gammaherpesvirinae* (EBV, KSHV) [4,5,6].

After viral entry, herpesvirus capsids are transported to the host cell nucleus, where they dock onto nuclear pores, and genomes are released into the nucleoplasm [7]. At this point, viral genomes can either start to replicate and initiate a lytic infection cycle or go latent, leading to a persistent infection that can reactivate later. Viral and host factors in the viral genome’s microenvironment play a critical role in this decision [8,9]. In both cases, viral proteins associate with the viral genomes in membrane-less compartments called replication compartments (RCs) for lytic replication and nuclear inclusions or speckles during latency [10]. However, the biophysical principles of their formation and properties have been largely unclear. Recent work has suggested that cellular membrane-less compartments are often biomolecular condensates formed by liquid–liquid phase separation (LLPS). Therefore, they are frequently referred to as liquid organelles. LLPS describes the separation of a mixture into two fluid phases as a function of their respective concentrations and the thermodynamic environment, e.g., temperature and pH. It occurs when interactive forces between groups of molecules outweigh the entropic forces (Figure 1A,B), causing these components to become locally concentrated in a separate phase. Demixing can be nucleated in the binodal regime while spontaneously occurring in the spinodal regime [11]. The concept of LLPS has recently been applied from soft matter physics to biological systems [12], describing membrane-less cellular organelles as biomolecular condensates [13,14], and was highly successful in describing the behavior of various main components of membrane-less compartments [15,16,17]. In general, biomolecular condensates seem to consist of two main classes of proteins: (1) Multivalent proteins constituting the building blocks of the condensate, often called scaffolds; (2) Lower valency molecules that bind to these scaffolds and can exchange dynamically called clients. Scaffolds can be proteins carrying intrinsically disordered regions (IDRs) and/or multimerization domains and nucleic acids [14,16,18,19]. Clients are typically binding to scaffolds via specific interaction motifs. IDRs mediate multivalent interactions through weak binding modes, including cation pi–pi, dipole–dipole, and cation–anion interactions. When scaffold proteins reach their saturation concentration (C_sat_), they self-associate by multivalent heterotypic interactions, cluster, and undergo LLPS [13,20] (spinodal regime). If the concentration is too low, but instead is in the binodal regime, nucleation of LLPS can be initiated by compatible seeds, such as RNA or DNA [13,21]. Nucleation is vital for the spacial regulation various processes, such as DNA repair, transcription, or epigenetic modification [22,23]. In general, nucleation in the binodal regime allows the cell to react to triggers by dynamically generating a non-linear response. It is important to note that almost every protein will phase separate at high concentrations [24]. In contrast, scaffold proteins do so at physiological concentrations while clients only associate with the phase-separated condensate when free binding sites exist. Scaffolds can be modified by post-translational modifications (PTMs), such as phosphorylation and SUMOylation, and IDRs are especially susceptible due to their molecular accessibility [25,26]. PTMs can reversibly alter IDR charge and hydrophobicity or change their structure [27,28,29,30,31,32,33]. Client recruitment is tuned by these modifications and by changing the scaffold’s composition, resulting in a highly dynamic structure that compartmentalizes selected molecules [14,27].

## 2. Evidence for Biomolecular Condensate Formation during Herpesvirus Latency and Lytic Replication

Some recent studies could show that condensate formation of key viral proteins plays an important role during herpesvirus latency. KSHV induces nuclear inclusions of its latency-associated nuclear antigen (LANA) to tether viral genomes to centrosomes during cell division. LANA binds to the terminal repeats (TR) of KSHV viral genomes, multimerizes, and contains an IDR. LANA oligomerization is required for the formation of nuclear bodies and the colocalization with important nuclear factors, such as DAXX, EZH1, and the Origin Recognition Complex (ORC) [34]. Indeed, LANA condensates are sensitive to 1,6-Hexanediol, and perturbating phase separation alters KSHV genome maintenance, but it is not inducing the viral lytic cycle [34]. This might be due to repressive chromatin markers, such as H3K27 trimethylation and H2AK119 ubiquitylation on the chromatinized genome [35,36]. During the KSHV replication cycle the genome becomes associated with activating chromatin markers, and the six core replication proteins are then expressed [34,37]. Interestingly, co-transfection of these six proteins was shown to be sufficient for forming replication compartments (RCs) [38]. Since only ORF59 and ORF6 harbor disordered regions according to AlphaFold2 structural predictions (www.bosse-lab.org/herpesfolds, accessed on 1 May 2022), these might be prime candidates for further studies. During lytic replication, LANA remains associated with KSHV viral genomes in ring-like structures; meanwhile, DAXX is evicted from the compartment [34]. This implies a major shift in the compartment’s composition with the transition from LANA-NBs to RCs. Whether phase separation plays a role in tuning their composition during the latent-to-lytic transition is not known. Similar is also true for the Epstein–Barr virus (EBV) EBNA2 protein, a transcription factor that modulates EBNA-LP, EBV latent gene transcription, and numerous cellular genes. Unlike LANA, it cannot bind DNA directly but needs other factors to interact with it [39]. It has recently been reported that both proteins exhibit large disordered regions that can form dynamic compartments at super-enhancers in transfected cells and in vitro [40,41]. Interestingly, FRAP experiments of both proteins do not show full recovery, suggesting that EBNA2 binding partners of these molecules play a role in their immobilization [40,41].

Concerning lytic herpesvirus replication and bona fide RCs, we recently found that the HCMV proteins UL112-113 undergo LLPS to create a pro-replicative environment around incoming genomes to ensure their replication [42]. Newly formed pre-replication compartments (pre-RCs) marked by UL112-113 appear spherical and recover sphericity after fusion, indicating surface tension. At early times post-infection, they also exhibit fast fluorescence recovery after photobleaching (FRAP) within seconds, indicating fluidity. Moreover, they are highly sensitive to inhibitors of weak molecular interactions, such as 1,6-Hexanediol or 1,2-Propanediol, indicating that condensate formation is mediated by weak interaction of the UL112-113 IDRs [42]. Clustering the UL112-113 IDRs is sufficient for the formation of liquid condensates. Moreover, UL112-113 condensates can be nucleated by exogenous DNA, and UL112-113 phase separation at viral genomes is essential for their replication. It is, therefore, likely that UL112-113 acts as a scaffold forming pre-RCs. Importantly, the essential viral polymerase processivity factor UL44 cannot form condensates on its own but is recruited as a client into pre-RCs by its IDR, thereby ensuring that replication factors are concentrated at viral genomes [42]. In contrast, a study aimed at analyzing why RNA Polymerase II (RNApolII) is sequestered out of cellular chromatin and enriched in HSV-1 RCs came to seemingly different conclusions [43]. McSwiggen and colleagues started with the hypothesis that RCs are phase-separated condensates in which RNApolII is enriched via its disordered C-terminal domain (CTD). FRAP analysis of PolII showed a higher immobilized fraction in infected cells than uninfected, which increased about 15–20% as infection progressed from 3–4 to 5–6 hpi. However, neither mutations of the CTD nor inhibiting weak interactions with 1,6-Hexanediol showed any effect on RNApolII sequestration. Next, they quantified the diffusion of RNApolII inside and outside of the RC area. They did not find any difference for the freely diffusing fraction, while more bound RNApolII was found inside the RC area. Evaluating the RNApolII jump length (i.e., the displacement between frames) in or out of the RC did not show any difference. The authors concluded that the HSV-1 RCs are not liquid condensates since RNApolII molecules should show divergent diffusion patterns when crossing the condensate interface. Instead, RNApolII could bind with a much higher affinity to viral genomes than to host chromatin because viral genomes are largely devoid of nucleosomes. The authors concluded that RNApolII is not sequestered into HSV-1 RCs by interacting with a so-far undescribed viral phase but because the unchromatinized viral genomes display more binding sites [43]. The enrichment of RNApolII at open viral genomes is an important reminder that other mechanisms than LLPS can mediate the enrichment of molecules. In our view, these results do not exclude that viral proteins involved in the formation of HSV-1 RCs undergo LLPS nor that the RC might be a biomolecular condensate. The aforementioned study only analyzed the mechanism of RNApolII sequestration into RCs but did not characterize the properties of viral proteins involved in RC biogenesis. Indeed, the same study found that viral proteins expressed with immediate-early kinetics, which are important for RC formation, are highly enriched in IDRs. Moreover, HSV-1 RCs can coalesce in vitro [44,45,46], indicating at least some potential fluidity. In addition, a recent study on ICP4 by Seyffert and colleagues concludes that the essential RC protein ICP4 of HSV-1, which acts as a viral transcription factor, forms biomolecular condensates in transfected and infected cells at viral genomes and confers liquid properties to HSV-1 RCs [47]. Since the authors used a viral protein for studying the properties of HSV-1 RCs, these data might be more relevant for studying the nature of HSV-1 RCs than solely analyzing the behavior of a cellular protein in the RC. If the HSV-1 RC indeed constitutes a phase-separated biomolecular condensate, the question is, why does RNApolII exhibit no change in diffusive behavior when crossing into the RC area? The first potential explanation is related to the viscosity of the HSV-1 ICP4 condensates: The Stokes–Einstein equation describes the diffusion of a particle undergoing Brownian motion. It defines the diffusion coefficient D as D = kBT/6πRη with kB being the Boltzmann constant, T the absolute temperature, R the radius of the particle, and η the viscosity of the medium. In a system with constant temperature, the diffusion coefficient of a particle with a fixed size is only dependent on the viscosity η of the medium in which it diffuses. Accordingly, the diffusion coefficient of RNApolII would only change if the viscosity of the RC is different than the nucleoplasm. Unfortunately, no data on HSV-1 RC viscosities are currently available. Using fluorescent lifetime imaging, the protein concentration in the nucleoplasm and nuclear speckles of HeLa cells was estimated at 150 mg/mL [48]. In contrast, the concentration of nucleoli was measured to be almost twice as much. Similar results were obtained for membrane-less organelles from Xenopus oocytes [49]. Both nuclear speckles, as well as nucleoli, show features of LLPS [50,51]. Therefore, biomolecular condensates, such as the RC and the nucleoplasm, can have similar protein concentrations. Assuming that the proteins in both the dilute and dense phases have similar properties, it is likely that the dense phase does not exhibit a higher viscosity to RNApolII than the surrounding nucleoplasm. Of course, this assumption depends on the specific properties of the concentrated molecules. Microrheology with inert probes in the same scale range as RNApolII or image correlation spectroscopy (ICS) are needed [52] to solve this question. A second potential and closely related explanation relates to the fact that biomolecular condensates seem to possess a mesh size that allows passive penetration of inert molecules, such as fluorescent dextrans up to a size of 2000 kDa, depending on the organelle and cell type [49,53]. This would allow corralled diffusion in and out of a potential biomolecular condensate. The extent of corralling depends on the ratio between particle diameter and the mesh size, as it was shown that small molecules are less subjected to corralling than larger complexes or macromolecular complexes, which can be trapped in chromatin corrals [52,54,55,56]. Again, ICS or microrheology of inert probes, such as fluorescent dextrans, could clarify this point. Notably, dextran permeability assays have already been used to determine the mesh size of LAF-1 droplets in living cells [53], and a similar approach could be applied to herpesvirus RCs.

## 3. Herpesvirus Replication Compartment Maturation

Although pre-RCs are likely homogenous biomolecular condensates, their properties change with ongoing viral infection. We found that HCMV RCs lose their spherical shape around 30 hpi, indicating that intra-RC interactions overcome surface tension at this point [42] (Figure 1C). Moreover, RCs expand dramatically during infection, at the end filling almost the complete host nucleus. Mature RCs are much more resistant to inhibitors of weak interaction, such as 1,6-Hexanediol or 1,2-Propanediol, indicating that stronger molecular interactions stabilize the condensate. Moreover, mature RCs, show less motility and slower recovery after photobleaching and a lower fraction of mobile molecules [42]. Importantly, PAA, an inhibitor of viral DNA replication, abrogates condensate maturation and reverses most of the changes, indicating that viral genomes likely play a role in the maturation of RCs as a condensate [42]. This fits our observation that the association of the viral polymerase processivity factor UL44 with pre-replication compartments, which marks the onset of DNA replication, leads to the observed phenotypic changes in the compartment’s properties [42]. These findings suggest a role for viral genomes in modulating RC condensate properties, likely by providing another scaffold or mesh to which the condensate´s proteins, such as the UL112-113 proteins, but potentially also other viral genome-associating proteins, such as the single-stranded DNA binding protein UL57 (an HSV-1 ICP8 homolog), can bind. Similar data were also reported for HSV-1 ICP4. In an elegant assay, the authors labeled viral replicons in living, infected cells and measured their FRAP recovery compared to the DNA-binding ICP4. They found that the protein fluorescence recovered several times faster than viral genome fluorescence, suggesting a role for viral DNA as an additional, more rigid scaffold for the condensate [47]. However, in contrast to HCMV UL112-113 proteins, recovery of ICP4 condensates did not change between transfected and infected cells. In addition, ICP4 condensates in transfected cells showed much less sphericity and were only partially dissolved by 1,6-Hexanediol. This might be due to binding of ICP4 to cellular DNA. However, in vitro assays will probably be needed to define the nature of ICP4 condensates in the absence of DNA. Importantly, such changes in the composition of the condensate will very likely also change its properties, such as pore size and its permeability for cellular factors. Therefore, one has to keep in mind that RC properties need to be quantified as a function of infection progression. Similar viscous to solid transitions of biomolecular condensates as observed with herpesviruses have also been described for cellular components in vitro and in vivo [24,57,58,59]. Some compartments can undergo phase transition through a process called hardening or maturation [14,18,59]. For example, increasing RNA concentrations alters Whi3 condensate viscosity [18]. Interestingly, the length of the added RNA molecules influences LAF-1 droplets, with short ones lowering and long ones increasing the apparent viscosity [53]. Moreover, telomeric DNA serves as a scaffold for the oligomerization of phase separating proteins at telomeres, and shelterin condensates exhibit gel-like properties in the presence of telomeric DNA in vitro [60]. Short RNA results in more liquid droplets, while longer RNA or DNA promotes more viscous droplets [15,18,60]. In addition to viral DNA providing a rigid mesh that hardens the RC, PTMs of scaffolds and clients likely also play a role in RC maturation. In particular, phosphorylation and SUMOylation are prominent during HCMV infection. HCMV encodes the viral serine/threonine kinase pUL97, which is important for the regulation of viral DNA synthesis, and modification of cellular transcriptional and translational factors [61]. UL97 phosphorylates several viral proteins, such as the polymerase processivity factor UL44 [62] and might play a role in condensate formation. Similar mechanisms were reported for rotaviruses [63,64]. The cellular SUMOylation pathway also plays a regulatory role in infection by many different viruses, including HCMV. Several RC proteins can be SUMOylated [63,64,65]. However, further studies are needed to define the role of PTMs in scaffold and client affinity and RC maturation. Finally, it is still very much unclear how subcompartmentalization of the RC is achieved since the mature RC is not a homogenous structure [66]. The replication of viral DNA occurs at its periphery, and the newly synthesized genome migrates to the center, with the ssDNA binding protein (ICP8 or UL57) and UL44 occupying different parts of it, indicating a spatial diversification of transcription and DNA replication [67]. We have also observed that late HCMV RCs show areas of different densities. Again, similar phenotypes are found in liquid organelles of uninfected cells. For instance, nucleoli can spontaneously rearrange into different sub-phases [51,68], and super-resolution microscopy of P-granule structures revealed a stable core substructure surrounded by a phase-separated shell, with the number of cores linearly correlating with stress granule volume [69]. Potentially, these concepts also apply to RCs, with stable DNA-protein patches acting as a core [32]. 

## 4. Conclusions

As outlined, cumulative data suggest that KSHV and EBV during latency and HSV-1 and HCMV during lytic replication form biomolecular condensates at viral genomes via LLPS. Further efforts are needed to expand these initial data to more herpesviruses. Importantly, herpesvirus condensates seem to change their properties and constituents throughout infection but more detailed studies are needed. It is likely that only the initial stages of RCs conform with more simplistic models of LLPS when nucleation of viral proteins at genomes leads to relatively homogenous and fluid condensates at viral genomes (Figure 1B). This mechanism might help in concentrating resources to maximize initial transcription and replication. With ongoing viral replication, increased concentration of viral DNA that acts as long molecular scaffold will cause molecular crowding inside the compartment, forcing disordered proteins to engage in more stable bonds with surrounding molecules and creating more complex entanglement (Figure 1C). Similar mechanisms have been shown for replication compartments of Adenovirus and Rotavirus viral factories, which show liquid properties when they arise and convert to a more viscous state with the progression of infection [64,70]. Right now, data are lacking on the properties of very late herpesvirus RCs. Herpesviruses exploit aggregate formation for evading the immune responses [71,72], and it seems possible that RCs maturing into aggregates might sequester cellular antiviral factors. Moreover, viral condensates might be valuable models to study the maturation or aging of biomolecular condensates as they represent relatively low complexity subjects with chronologically well-defined progression. 

Finally, a recent landmark study showed that respiratory syncytial virus (RSV) replication can be inhibited by a compound that induces the hardening of inclusion bodies, the RNA-virus analog of herpesvirus RCs [73], although the structural mechanism remains unknown. Future research is needed to better define the complex molecular interactions that mediate viral condensates. Uncovering these principles might ultimately lead to a new class of potentially broadly-acting viral inhibitors. 

## Figures and Tables

**Figure 1 viruses-14-00960-f001:**
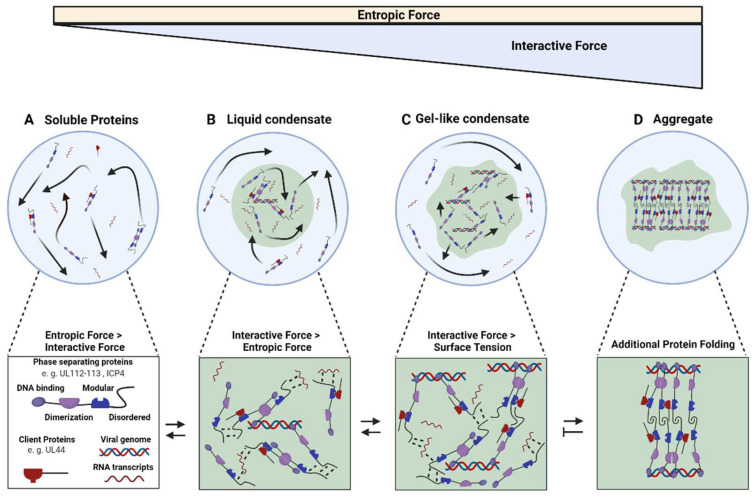
Model of the molecular interactions during herpesvirus replication compartment formation and maturation. (**A**) At constant temperature, molecules in a solution move at a fixed rate via Brownian motion leading to a defined level of entropic force. (**B**) When interactive forces overcome these entropic forces, for example, by increasing the concentration of a molecule, demixing occurs. Since keeping an interface between the two phases is energetically expensive, surface tension forces the fluid compartment into a sphere, maximizing the volume per surface area. These liquid condensates are characterized by high internal molecular diffusion and selective external diffusion. (**C**) Increasing the concentration of components and adding in molecules with higher valency, such as long RNA or DNA molecules, induces the formation of more bonds, finally overcoming surface tension resulting in an irregularly shaped condensate. (**D**) Increased bond stability can finally force disordered regions to fold and entangle in new interactions, further increasing molecular crowding and kinetically arresting the involved molecules. These aggregates might not be reversible. Note that aggregate formation has not been described for herpesviruses replication compartments, yet.

## Data Availability

Not applicable.

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
