# Peer review of "Herpesvirus Replication Compartments: Dynamic Biomolecular Condensates?"

_viruses, 2022, doi:10.3390/v14050960_

Round 1

Reviewer 1 Report

The authors discuss the hypothesis that HSV-1 replication compartments (RC) may represent condensates into which RNAPolII partitions. They argue that mutations in the C-terminal domain of the RNAPolII or the use of 1,6-hexanediol did not have an effect on RNAPolII. Moreover, the diffusion pattern of the RNAPolII in and out of the RC suggested that the RC are not liquid condensates. Despite these published data, the authors conclude that RCs may still represent condensates since the data aren’t fully conclusive to substantiate the notion that RCs are not condensates. They do however raise the question of why RNAPolII does not exhibit the behavior typical for molecules that interact with the liquid phase of RCs. They argue that the diffusional behavior of RNAPolII inside and outside the RCs could be similar given that the viscosity of the dense phase of the RCs may be comparable to that of outside the RCs (dilute phase). They suggest that microrheology with particles may be required to solve this dispute, however, it should be noted that these experiments can only be reliably performed in vitro. I would suggest the authors also mention other in vivo techniques (FRAP for RNAPolII or image correlation spectroscopy instead of simply using frame-by-frame analysis of RNAPollI motion).

Major comments:

  1. Perhaps one major issue is that the authors suggest that RNAPollI has a higher affinity for the viral DNA, implying that RNAPolII must have longer dwell times within the RCs. However, they also suggest that RNAPolII does not diffuse slower within the RC. Even with RCs representing a mesh within which RNAPol could diffuse ‘freely’, it is unlikely that RNAPol would diffuse freely in this case, since such a mesh would also have an effect on the diffusion (corralled diffusion as opposed to the free 3D diffusion model). I think the authors need to carefully reconsider their model, and try to reconcile it with the existing experimental data. Perhaps, RCs indeed represent condensates, and there’s something fundamentally wrong with the previously published data on the motion of the RNAPoll within and outside RCs. Perhaps, the authors looked only at certain types of RCs that underwent further maturation/gelation? The authors do discuss this point, and similar observations have been made for other viral condensates, e.g., in replication factories of rotaviruses, for which 1,6HD and 1,2PD were also used to discriminate between the early infection (liquid-like) and the mature condensates.
  2. I think the manuscript would be improved by adding another figure/model of the RC condensate, as currently Figure 1 describes more generic principles of condensate formation, whereas this manuscript specifically describes the hypothetical model of the condensates representing HSV-1 RCs.

Author Response

We thank both reviewers for critically reading our manuscript. Below you find our point-by-point response to the reviewers’ suggestions highlighted in light blue.

Reviewer #1

The authors discuss the hypothesis that HSV-1 replication compartments (RC) may represent condensates into which RNAPolII partitions. They argue that mutations in the C-terminal domain of the RNAPolII or the use of 1,6-hexanediol did not have an effect on RNAPolII. Moreover, the diffusion pattern of the RNAPolII in and out of the RC suggested that the RC are not liquid condensates. Despite these published data, the authors conclude that RCs may still represent condensates since the data aren’t fully conclusive to substantiate the notion that RCs are not condensates. They do however raise the question of why RNAPolII does not exhibit the behavior typical for molecules that interact with the liquid phase of RCs. They argue that the diffusional behavior of RNAPolII inside and outside the RCs could be similar given that the viscosity of the dense phase of the RCs may be comparable to that of outside the RCs (dilute phase). They suggest that microrheology with particles may be required to solve this dispute, however, it should be noted that these experiments can only be reliably performed in vitro. I would suggest the authors also mention other in vivo techniques (FRAP for RNAPolII or image correlation spectroscopy instead of simply using frame-by-frame analysis of RNAPollI motion).

We thank Reviewer #1 for his suggestions. We now discuss the FRAP data of PolII performed by Mcswiggen et al. (lines 152-154). Furthermore, we added ICS (line 252 and line 259) as a further experimental avenue to characterize the properties of HSV-1 RCs. Moreover, we added a reference where dextran permeability has been used in living cells to explore the properties of biomolecular condensates (lines 260-262).

Major comments:

Perhaps one major issue is that the authors suggest that RNAPollI has a higher affinity for the viral DNA, implying that RNAPolII must have longer dwell times within the RCs.

This was a conclusion of the original work and not our suggestion.

However, they also suggest that RNAPolII does not diffuse slower within the RC. Even with RCs representing a mesh within which RNAPol could diffuse ‘freely’, it is unlikely that RNAPol would diffuse freely in this case, since such a mesh would also have an effect on the diffusion (corralled diffusion as opposed to the free 3D diffusion model).  I think the authors need to carefully reconsider their model, and try to reconcile it with the existing experimental data.

We absolutely agree with Reviewer #1 that a mesh-like structure would lead to corralled diffusion. The extent of corralling, however, depends on the ratio between particle diameter and mesh size as it has been shown that small molecules such as soluble GFP “experience” much less corralling in chromatin than larger molecules or macromolecular complexes such as viral particles or microinjected beads which are essentially trapped in chromatin corrals (Görisch 2005, Bosse 2015, Görisch 2004, Wachsmuth 2000). Therefore, RNAPolII diffusive behavior while crossing the condensate border will be likely determined by the ratio of mesh size and/or viscosity between the condensed and the dilute phase. If mesh size and viscosity of both phases are similar, little difference in the diffusive behavior of RNAPolII will be measurable. Since the authors of the original publication did not measure these properties, no conclusion can be drawn about the condensate properties of HSV-1 RCs. For this reason, microrheology, FRAP, and ICS as suggested by reviewer #1 are needed. We wanted to make exactly this point in our commentary and modified the text accordingly (lines 252-262).

Perhaps, RCs indeed represent condensates, and there’s something fundamentally wrong with the previously published data on the motion of the RNAPoll within and outside RCs. Perhaps, the authors looked only at certain types of RCs that underwent further maturation/gelation? The authors do discuss this point, and similar observations have been made for other viral condensates, e.g., in replication factories of rotaviruses, for which 1,6HD and 1,2PD were also used to discriminate between the early infection (liquid-like) and the mature condensates. I think the manuscript would be improved by adding another figure/model of the RC condensate, as currently Figure 1 describes more generic principles of condensate formation, whereas this manuscript specifically describes the hypothetical model of the condensates representing HSV-1 RCs.

We thank reviewer #1 for his thoughts and modified Figure 1 to better reflect the data published on herpesvirus RCs by inserting viral example factors.

Reviewer 2 Report

This commentary by Enrico et al. discusses the role of biomolecular condensates mediated by liquid-liquid phase separation on herpesvirus biology. Based on their recent observations on HCMV and other literature available, they describe the structure, composition, and dynamics of herpesvirus replication compartments (RCs). This is a well-written manuscript on a relevant and important topic.

  • My main criticism of this manuscript is that this commentary is primarily focused on HCMV and not all herpesviruses have been described with equal weightage. In my opinion, based on the title – “Herpesvirus replication compartments:”, there is scope and literature available to include more information on RCs of other herpesviruses in detail.
  • Information on the difference between latent and lytic phases of herpesvirus RCs is missing and can add more value to the present manuscript. The authors should consider including more information on this topic.
  • Typos: Line 197 – ‘whi’, word missing? Line 32 – parenthesis not closed.

Author Response

We thank both reviewers for critically reading our manuscript. Below you find our point-by-point response to the reviewers’ suggestions highlighted in light blue.

Reviewer #2

This commentary by Enrico et al. discusses the role of biomolecular condensates mediated by liquid-liquid phase separation on herpesvirus biology. Based on their recent observations on HCMV and other literature available, they describe the structure, composition, and dynamics of herpesvirus replication compartments (RCs). This is a well-written manuscript on a relevant and important topic.

We thank reviewer #2 for his kind support of our manuscript.

My main criticism of this manuscript is that this commentary is primarily focused on HCMV and not all herpesviruses have been described with equal weightage. In my opinion, based on the title – “Herpesvirus replication compartments:”, there is scope and literature available to include more information on RCs of other herpesviruses in detail.

Reviewer #2 is correct that there is a large body of literature dealing with herpesvirus RCs. Unfortunately, there is very little known about the condensate properties of herpesvirus RCs except the literature that we discuss. We modified the text to highlight this limitation by adding the need for future research to the outlook (line 256-257).

Information on the difference between latent and lytic phases of herpesvirus RCs is missing and can add more value to the present manuscript. The authors should consider including more information on this topic.

We thank Reviewer #2 for this suggestion and expanded the section on the role of condensates during herpesvirus latency (see lines 79-135). However, there are only three studies published on the role of condensates in herpesvirus latency which we already cited in the previous version of the manuscript (Vladimirova 2021, Peng 2020, Yang 2021).

Typos: Line 197 – ‘whi’, word missing? Line 32 – parenthesis not closed.

We corrected the typos and did another round of spell-checking (line 32 and 329).

Round 2

Reviewer 2 Report

The Author's rebuttal is satisfactory and this manuscript can be accepted for publication.